# The Lived Self-Care Experiences of Patients Undergoing Long-Term Haemodialysis: A Phenomenological Study

**DOI:** 10.3390/ijerph20064690

**Published:** 2023-03-07

**Authors:** Sisook Kim, Hyunsook Zin Lee

**Affiliations:** 1Department of Nursing, Hwasung Medi-Science University, Hwaseong-si 18274, Republic of Korea; 2College of Nursing, Kyungdong University, Wonju 24695, Republic of Korea

**Keywords:** self-care, long-term care, renal dialysis, haemodialysis, patients, qualitative research, phenomenological study

## Abstract

The study aims to understand the lived self-care experiences of patients who have undergone long-term haemodialysis. The study adopts a qualitative phenomenological design. Data were collected for six months, from 1 July to 31 December 2020. Out of 90 outpatients in a haemodialysis clinic at a university hospital in Seoul, Korea, 11 patients who had received haemodialysis for more than 10 years were purposefully selected, and 9 of them took part in in-depth interviews. The main research question was, ‘What was the experience of surviving long-term haemodialysis?’ The study revealed four main themes surrounding the topic of self-care: (A) the desire to keep living despite challenges, (B) creating one’s own dietary principles, (C) moving one’s body with the remaining strength, and (D) moving toward independence. In the long-term self-care of haemodialysis patients, they shared personal observations on their disease and treatment process and their struggles to try to manage their own physical and emotional self-care. By exploring the experience of long-term haemodialysis, it is possible to gain a deeper understanding of their perceptions, emotions, and motivations. With this information, healthcare professionals can develop interventions and support strategies that are tailored to the specific needs of haemodialysis patients.

## 1. Introduction

According to Orem’s self-care theory [1], self-care is an intentional act and the process of making and executing judgments on a person’s regulatory requirements. Parts of the process of self-care may be separated by time and space in terms of a person’s state of human functioning and human development [2]. However, most self-care practices for haemodialysis patients are concentrated on physical care, such as monitoring dialysis vessels or controlling diet, fluid intake, and weight [3,4]. Although haemodialysis patients suffer from symptom mitigation and deterioration, similar to other patients with chronic illnesses, the formal cycle of acceptance, adaption, and regaining control may not be sufficient to understand these patients, as has been explored in qualitative studies [5].

For example, to prevent electrolyte imbalance between haemodialysis treatments, water and food intake should be carefully monitored and consumed. However, excessive dietary control in haemodialysis patients can lead to malnutrition, and such dietary restrictions and nutritional deficiencies in haemodialysis patients are predictors of early mortality [6]. Although there is a continuous need to review the criteria and recommendations for dietary restraint in hemodialysis patients to maintain good nutritional status [7], it is still not easy to consider specific individual characteristics such as gender, race, genetics, and sociocultural environment.

There are also differences or conflicts in the self-care of haemodialysis patients. Hemodialysis patients consider self-management to be a balance between lifestyle change, emotional health, and medical treatment, whereas medical practitioners have emphasised the aspect of compliance with treatment guidelines [8]. Nevertheless, healthcare providers are still struggling with how to adhere to haemodialysis patients [9,10,11]. Studies have reported on the caregiver burden of haemodialysis patients [12,13]. Haemodialysis patients already know this and often worry about what burden they may place in unavoidable situations between their family and friends and needing care [14]. As such, the care of haemodialysis patients has paradoxical aspects and requires an integrated perspective that takes into account the characteristics of chronic and severe diseases, the difficulty of treatment guideline implementation, and the balance between life and treatment.

Although there is a large body of literature regarding the best practices of self-care for haemodialysis patients [3,5,6], the care of haemodialysis patients has gaps and ambiguity in the knowledge in this field. This study aims to rigorously and transparently explore the experience of patients who endure long-term haemodialysis. A phenomenological method was used to obtain theoretical and practical findings through a practical and stepped method [15]. By exploring the subjective experience of people with long-term haemodialysis, it is possible to gain a deeper understanding of their perceptions, emotions, and motivations. This can inform the development of interventions and support strategies that are tailed to the specific needs of these individuals, taking into account their unique perspectives and experience.

## 2. Materials and Methods

### 2.1. Study Design

This qualitative study used phenomenological methods to explore the experience of self-care in patients who received long-term haemodialysis.

### 2.2. Participants

Participants in this study included patients aged 19 years or older who had received haemodialysis for more than 10 years with no kidney transplantation or peritoneal dialysis. Patients with mental illnesses or cognitive impairment and inpatients were excluded. Participants were selected from a group of 90 outpatients at a haemodialysis clinic at a university hospital in Seoul, Korea. Eleven patients who received haemodialysis for more than 10 years were purposefully selected, and nine of them participated in this study (see Figure 1 and Table 1).

### 2.3. Data Collection

Data were collected for six months, from 1 July to 31 December 2020. The interview was held in a seminar room located in the haemodialysis clinic before participants initiated their haemodialysis session. Each participant was interviewed at least twice by the researchers, with the interviews lasting 50 to 80 min. When additional questions and answers were required, supplementary interviews of no longer than 30 min were administered at the patient’s bedside or the seminar room, as the patient’s condition allowed. Each interview was recorded and transcribed with the consent of the participants.

### 2.4. Data Analysis

Data were analysed using a phenomenological approach [16]. First, the researchers read the entire transcript repeatedly until they could understand the general scope of the data. At this stage, the recordings were separated and labelled to avoid admixing the experiences of individual participants. In the second step, researchers read the transcripts and classified each statement into a specific category. The most relevant statements were reviewed multiple times. Direct and psychologically more sensitive expressions were carefully inspected, and essential statements were extracted with the help of free imaginative variation. In the third step, which was the core part of the analysis, the researchers transformed the data. The majority of what the participants said was retained, but focus was given to their psychological underpinnings. The psychologically relevant expressions were then examined multiple times. An overview of these steps is illustrated in Table 2.

**Figure 1 ijerph-20-04690-f001:**
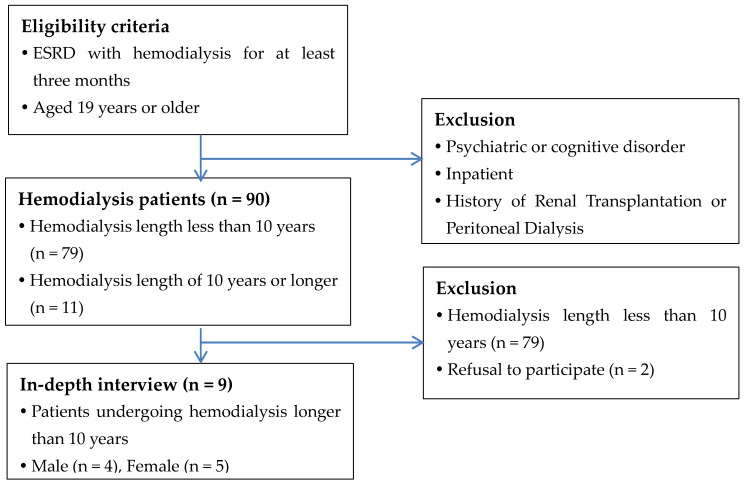
Inclusion and exclusion criteria of participants.

**Table 1 ijerph-20-04690-t001:** Participant demographics and information.

Participant ID	Gender	Age (Years)	Hemodialysis Period	Causes of Disease	Family Living Together with Participant
P1	Female	74	10 year 7 month	Diabetes Mellitus	Son, granddaughters
P2	Female	72	11 year 5 month	Hypertension	Husband
P3	Male	75	10 year 4 month	Unknown	Wife
P4	Male	60	18 year 10 month	Hypertension	None
P5	Male	69	13 year 6 month	Hypertension	Wife, son, daughter
P6	Male	62	10 year 8 month	Hypertension	Wife
P7	Female	66	10 year 5 month	Diabetes Mellitus	None
P8	Female	51	16 year 7 month	Hypertension	Husband
P9	Female	62	16 year 4 month	Unknown	Daughter

**Table 2 ijerph-20-04690-t002:** Example of the step-wise qualitative analysis process.

Participant	First Step	Second Step	Third Step
P8	“…[During hemodialysis] as I watch television, close my eyes, and think this and that…. those thoughts of “Why should I live like this?” and “What did I do wrong?” come to mind. Sometimes, I lie down and cry. But rather than people suffering from cancer and chemotherapy, it’s still possible to live on hemodialysis, so I got this kind of comfort…Let’s just be grateful that I can live on dialysis like this.”	“…[During haemodialysis] As I watch television, close my eyes, and think this and that…./ Those thoughts of “Why should I live like this?” and “What did I do wrong?” come to mind./ Sometimes, I lie down and cry./ But rather than people suffering from cancer, chemotherapy, it’s still possible to live on hemodialysis, so I got this kind of comfort…/ Let’s just be grateful that I can live on dialysis like this.”	Trying to find the cause of the disease in himself. Hopeless about his situation on haemodialysis. Comforting and persuading himself with comparative advantages. Thinking it is more important to stay alive.

### 2.5. Validity and Reliability

The researchers maintained the rigour of the study by ensuring that the research methods and analytical findings were accurate, replicable, applicable, and free from bias. Data collected in the study were reviewed by all participants to ensure accuracy. The researchers also received feedback from experienced qualitative researchers regarding their analysis.

### 2.6. Ethical Considerations

Ethical approval was obtained from the institutional board of the hospital where the patients were treated. All participation was voluntary, and the required written consent of participants was obtained. Participants were informed that they could withdraw from the study at any time and that their personal information would be kept confidential. After the interviews, participants were given a water bottle with a scale for controlling fluid intake as an expression of appreciation for their help with the study.

## 3. Results

### 3.1. Participants’ Characteristics

The characteristics of the participants are listed in Table 1. Of the nine participants, four were men and five were women. The dialysis treatment duration of participants ranged from 10 years and 4 months to 18 years and 10 months, with an average duration of treatment of 13 years and 9 months. The average age of the participants (aged 52 to 75 years) was 65.7 years, and the age at which they received their first haemodialysis treatment ranged from 35 to 65 years. The primary cause of the disease was reported as high blood pressure for five patients and diabetes for two patients. For two patients, the cause was unknown. Most participants lived with their families, but Participant 4 had lived alone for several years since his mother’s death. Participant 7 lived with her son and daughter-in-law but recently began living alone.

### 3.2. Themes

The results of the data analysis focus on the process by which the haemodialysis patients cared for their health as well as their attitude toward their treatment. The analysis revealed four main themes: (A) the desire to keep living despite challenges, (B) creating one’s own dietary principles, (C) moving one’s body with the remaining strength, and (D) moving toward independence. Table 3 lists each theme and gives an overview of what each theme includes.

#### 3.2.1. Theme 1. The Desire to Keep Living despite Challenges

Participants reported feeling angry about being on haemodialysis but yet were thankful for the opportunity to have haemodialysis because it allowed them to survive. Although they were often ashamed of having to rely on haemodialysis, they expressed their distinct desire to keep living. As soon as they were connected to the dialysis machine, the sense of physical restraint became strong. Participant 9 experienced a loss of control and felt she was hanging upside down because the medical staff raised her legs to combat sudden low blood pressure. Participant 5 was frustrated because he had defecated in his underwear. He did this involuntarily as he sat down while experiencing severe dizziness. Participant 3 reported that his family often treated him as if he should be grateful to be alive. Participants comforted themselves with the knowledge that their current situations were less painful than chemotherapy or were more treatable than a rare disease with an unknown remedy. They often used a sense of humour to alleviate the pain of their disease.


*‘I have more thoughts of living now…I have to live 80 years old, don’t I? But I do not even know if I am 100 years old (laughing). When we are all at the clinic, oh, no, we are all gathered again to live like this (laughing). We say this: as we receive a bundle of medicine, tons of medicine, we say that we have a lot of precious medicine that is good for the body.’*
(Participant 1)

#### 3.2.2. Theme 2. Creating One’s Own Dietary Principles

Strategies for eating fewer fruits rich in potassium and green vegetables and limitations on fluid intake differed between individuals. When beginning haemodialysis, the participants reported that they felt helpless and had difficulty accepting their prognosis; however, they often countered this helplessness by learning about their disease and potential treatments. Many participants checked the correctness of their understanding with their doctors and nurses, often asking questions about the restricted food list and recipes while they were in the haemodialysis clinic. They focused on maintaining good relationships with hospital staff and other haemodialysis patients in order to have opportunities to discuss their diet. Nevertheless, the patients were not entirely trusting of healthcare providers. They often evaluated whether their healthcare providers’ advice was helpful or appropriate for them. As an alternative to the advice from healthcare providers, participants also received dietary advice from fellow dialysis patients as well as their caregivers. They met these patients and caregivers at the dialysis clinic and also met some of them through social media and the internet. Using information from all these sources, the participants monitored the food list they consumed, their clinical test results, and changes in their bodies to create their own unique dietary plans.

Participant 5 stated that he considered his body a water container that could only hold but never release water. He said that he drank water gradually, only when he really craved it, and slowly using a thin straw. When the water was in his mouth, he would hold it before spitting it back out. He would then repeat this process, never swallowing the water. Participant 9 reported that she wanted to drink iced coffee in the hot summer but replaced it with coffee-flavoured dark chocolate. She also stated that she used a small plate, similar to those she was given as part of a therapeutic diet that she had followed while in the hospital, and deliberately used chopsticks instead of a spoon in order to decrease her food intake.


*‘I used to mostly eat at restaurants on my way home after haemodialysis; however, this was not very good for me. (After eating out) I craved water. My blood pressure increased, and then I needed to take pills to control my blood pressure. So, I decided to cook and eat at home as much as I can, even if it is hard.’*
(Participant 8)

#### 3.2.3. Theme 3. Moving One’s Body with Remaining Strength

None of the participants reported that they had fully adapted to haemodialysis. They often barely made it to the clinic and wept in sorrow when they began haemodialysis because they felt like it was a punishment. Participants reported that they were so dizzy that they could not think and felt such weakness in their legs that they could not return home for hours after treatment. They suffered from unresolved itching and scratched their skin until it became purple. Participants visited the emergency room several times due to repeated intestinal bleeding, and then they learned that the cause of their pain was actually cardiovascular after they were transferred to an intensive care unit for cardiopulmonary resuscitation.

At certain points, despite the pain, participants felt that they had the power to be physically active. Even when feeling exhausted after treatment, they sensed that they had energy and moved their body at every opportunity. Participant 5 lost nearly 20 kg in the first two to three years of his treatment and felt he had lost all of his energy. Boredom and a desire to leave his home led him to start exercising, and eventually, he came to grow vegetables on his farm. He found that his body relaxed, and his strength came back when he would rest in the shade after being out in the sun in the fields. Participant 7 had to travel by bus to the haemodialysis clinic, and sometimes she had to sprint to catch the bus on the way home. She reported feeling lighter and less tired because she filtered out the scraps after haemodialysis.


*‘I’ve been exercising ever since I’ve been sick… for my body. I do not overexert myself, and I feel better little by little doing it like that. I feel a certain amount of lively energy.’*
(Participant 6)

#### 3.2.4. Theme 4. Moving toward Independence

Although long-term haemodialysis led to a dependency on family members and medical staff, low to relatively high levels of independence were also possible, depending on physical recovery. On exhausting days, participants moved their bodies slowly, taking a break without exertion while caring for themselves. They could finally recognise the principles of body operation by repeatedly moving their bodies and engaging in exercise for muscle strengthening. By the time they adjusted to the daily routine of haemodialysis, following physical recovery, they were able to step back from the conflict of surviving pain or the fear of death and could introspect about their life. They lost their jobs or moved away from their friends while visiting clinics for haemodialysis and were repeatedly hospitalised due to complications. Their family members were also busy living their own lives, and they could no longer ask for help with the excuse of endless dialysis. Although they were angry or resentful towards family members or their spouses for the aforementioned reason, they considered themselves as separate persons in their relationships with others. It was helpful in calming their emotions down and helping them to be free from the dependencies.

Participant 5, who started farming as an exercise, harvested crops such as sweet potatoes, peppers, and tomatoes and gave them to people in his neighbourhood every year because he knew that eating too much vegetables was harmful to him. Participant 9 stopped her dancing activities and took care of her granddaughter, driving her to kindergarten instead of having her working daughter do it. Participant 6 set a goal to go on his own to the haemodialysis clinic for the next 10 years, with the thought that if he received too much support, he could not insist on his rights in the family. Participant 2 thoroughly kept her daily routines, such as three meals a day, taking a walk, and exercising at a fixed time since she initiated haemodialysis, though it was difficult for her due to a low level of energy.


*‘This is not a curable disease. I am just maintaining life. If I do not get haemodialysis, my body swells up, and it is hard. As I have experienced, I try to overcome that… I can do it with my own strength. I try to do things myself. Even though I get support from others, [being dependent] is not going to help me or cure my disease.’*
(Participant 3)

## 4. Discussion

Physical self-care

The long-term hemodialysis patients struggled to survive even as they resented the lifestyle-limiting hemodialysis process. When patients felt even the slightest energy, they tried to use it to move their bodies. They moved their body with a desire to survive. Indeed, moderately vigorous physical activity, such as riding a mini bike or resistance training, can prevent fatigue build-up and improve physical performance in hemodialysis patients [17,18]. Additionally, although they had had more than a decade of hemodialysis, they were still independent and willing to take care of themselves. Instrumental activities of daily living (IADLs), which include food preparation, basic household chores, shopping, laundry, taking medication, using transportation and telephones, and managing expenses, require a higher level of autonomy and interaction with the environment than basic activities and are related to the quality of life in hemodialysis patients [19]. It is unclear whether physical activity or self-care are the secrets of long-term survival over 10 years, but at least these functional activities of daily living may be patients’ goals for remaining independently active.

The long-term haemodialysis patients’ self-care activities included careful observation of their disease and treatment, as well as creating and maintaining their own dietary principles. In this study, by changing and monitoring what they ate, the participants were able to better understand the relationship between their body and food and fluid intake, allowing them to establish their own diet principles. To cope with hunger and thirst, participants employed cognitive strategies, such as ‘avoiding certain substances, tasting food or water without swallowing, controlling one’s speed of eating, and thinking in advance about what consuming certain things could cause’ They also used behavioural strategies to change their habits. For example, participants reported substituting food with sweet and sour candy to increase salivation and avoiding spicy food and outdoor activities to suppress their thirst [10]. Considering that chronic excessive fluid intake is strongly associated with hypertension and mortality in patients undergoing haemodialysis, minimising the oral intake of fluid and salt using strategies such as choosing appropriate ingredients, excluding prohibited food, changing methods of cooking, and proportioning meals appropriately is important for the care of haemodialysis patients [6,10]. Previous studies have said that time, convenience, and financial constraints may hinder the dietary adherence of hemodialysis patients [10,11,20]. Patients who survived long-term haemodialysis believed empirically that the harder they moved their body, the more the energy to move came from them. On the other hand, most haemodialysis patients walk less than 5000 steps, and low fatigue has been reported in patients who walk more [21]. Fatigue is a subjective symptom. Haemodialysis patients’ fatigue is related to mortality and quality of life, but not much is known [22], and this does not necessarily fit with individual resilience, coping, control, and adaptation processes [23]. Further research on haemodialysis patients’ activity and fatigue is needed.

Many healthcare providers educate hemodialysis patients, but on the other hand, they need to learn from haemodialysis patients’ experience with fluid restriction and diet to overcome these obstacles [8,10].

Emotional self-care

Participants reported mixed feelings regarding the extension of their lives. Swedish haemodialysis patients reported being grateful but bitter for being able to stay alive despite the uncertainty and limited lifespan. They also reported feeling grateful for being able to remember what is important in life, such as family and friends [8,14]. Chinese haemodialysis patients placed value on having personal freedom and living in the moment rather than focusing on endless haemodialysis treatments and future despair [24], while Thai haemodialysis patients were satisfied with replacing their social life with family relationships during treatment [25]. Despite patients’ feelings of resentment toward their treatment, they were compelled to accept the changes in their lives and maintain their will to live because it was their only option.

However, it is noteworthy that there was little mention of hope among the participants in this study. Hopes of hemodialysis patients vary according to the age of the study group and study tools, but hopes may be important because they may be related to spiritual health [26,27,28]. In this study, the feelings of resentment toward treatment and a longing for a life extended by haemodialysis were opposing experiences that can be distinguished from reconciliation or hope for a new life. Although participants could maintain stability by enduring haemodialysis, they knew their survival was precious and limited. Similar to this, hope in the elderly with chronic heart failure was a belief in something positive without any guaranteed expectation that it would occur [29]. For participants who do not have a chance for a kidney transplant in the future, continuing to receive hemodialysis without further complications can be the best hope. While experiencing higher levels of socio-psychological stress and diverse feelings, such as anger, fear, depression, anxiety, and isolation, compared to physiological factors, patients used coping strategies to focus on their emotions, such as positive reappraisal [30]. During long-term haemodialysis, participants sought physical and mental independence and chose the strategy that worked most effectively to perceive themselves as objects lying at the boundary of modern medical science and fate. Hemodialysis patients have dealt with a variety of methods to reduce stress or despair, and, therefore, we need to reflect on the patients’ experience and whether mental and emotional interventions by healthcare providers are effective or detrimental [20,31].

It is important to note that this study’s approach is different from most qualitative studies on haemodialysis patients, which have focused on describing difficulties in health management or daily life. By investigating how patients approach and solve these difficulties, this study provides valuable insights into the self-care management strategies of haemodialysis patients. This phenomenological study is a useful approach in healthcare research as it can shed light on patients’ experiences that may be overlooked due to a lack of understanding and empathy in the healthcare system. It can help bridge the gap between the subjective experience of patients and the objective perspective of healthcare providers [11,13].

However, the study’s limitation of a small number of outpatient dialysis patients who have lived for a long time and the possibility of bias due to the experiences of Korean haemodialysis patients should be taken into account when interpreting the result. Future research should aim to replicate and expand upon these findings in larger and more diverse populations to increase the generalisability of the result.

## 5. Conclusions

The study of the lived experiences of long-term haemodialysis patients through integrative phenomenological methods extends our understanding of the nature of self-care. Long-term self-care of haemodialysis patients involves a complex and delicate balance of physical, emotional, and psychological observations and adjustments. Haemodialysis patients must constantly monitor their health, adhere to strict dietary guidelines, and undergo frequent medical procedures. This requires a great deal of discipline, resilience, and determination. However, self-care is not just about physical independence. It also involves developing a sense of inner strength and resilience that allows haemodialysis patients to cope with the challenges of their condition. This includes developing a positive attitude, finding support from friends and family, and learning to manage stress and anxiety. For many haemodialysis patients, the process of self-care becomes a way of life. They learn to prioritise their health and well-being and make adjustments to their lifestyles and daily routines in order to manage their condition and learn to find joy and meaning in their lives despite the challenges they face.

The experience of self-care in haemodialysis patients can be used as an example for the future planning of haemodialysis patients’ care and can be presented as evidence to guide other long-term healthcare strategies.

## 6. Strengths and Limitations

Although most qualitative studies of haemodialysis patients describe their medical care or the difficulties they face, the current study approaches an inadequately studied and crucial aspect of life for this population: self-care. One limitation of this study lies in the low survival rates of outpatient haemodialysis patients, as it was difficult to find a large sample of participants. The current study also presents only the experiences of Korean haemodialysis patients. This may introduce some level of bias, so the results may not be generalisable to a larger population.

## Figures and Tables

**Table 3 ijerph-20-04690-t003:** Four themes of self-care in patients undergoing long-term haemodialysis.

Main Themes	The Desire to Keep Living Despite Challenges	Creating One’s Own Dietary Principles	Moving One’s Body with Remaining Strength	Moving toward Independence
Meaning Unit	Enduring extended life with haemodialysis Being grateful for survival Having a desire for life to be extended	Understanding disease and treatment Experiencing the relationship between diet and the physical condition of the body Adjusting to a diet of one’s own	Detecting a streak of energy in extreme fatigue Snatching a ray of energy and moving one’s body Understanding the principle of energy flow inside the body	Attempting to be independent through physical recovery Realising the reality and waking up mentally. Focusing on oneself and taking care of others

## Data Availability

The datasets generated and/or analysed during the current study are not publicly available due to the inclusion of private information but may be available from the corresponding author upon reasonable request.

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
