# Peer review of "The Lived Self-Care Experiences of Patients Undergoing Long-Term Haemodialysis: A Phenomenological Study"

_ijerph, 2023, doi:10.3390/ijerph20064690_

Round 1

Reviewer 1 Report

Dear,, It  is  good  study  involved  ((The lived self-care experiences of patients undergoing long-term haemodialysis: A phenomenological study )),, but  to  accept  it  ,,  it needs   Minor  Corrections :

1- The  abstract  needs  more  clarification  for  results 

2- the  disccusion needs  explanation  of  results  ..

3- also  the  conclusions  are  not  clear ,, please add  more explanation  for  conclusions 

4- add  updated  references  ( 2023   and  2022)

5- The  results  need  to  more  citation  by  references  ,,

6- I accepted paper  after  Minor  Corrections  

Author Response

Thanks for your constructive review for my paper. here are the answer for your comments. 

Author Response

Dear Reviewer

Thank you for your constructive review for the paper. Here is the feedback for your comments.

Thanks and regards. 

Reviewer 3 Report

Psychological and lifestyle aspects of long-term hemodialysis (HD) are extremely important and sometimes underestimated by clinicians, who are mainly focused on patients’ compliance and treatment outcomes. The authors made an attempt to specifically analyze self-care experiences of extremely long-term HD survivors, which could help to improve our understanding of non-medical factors prolonging life of HD patients.

The presented article is interesting and carefully prepared, therefore, I suggest it to be accepted for publication after some revisions:

1) It is well known that socioeconomic factors could make a huge impact on the mental experiences of HD patients, including possibility to travel and work part-time (this is almost impossible for manual jobs), diet variety etc. This aspect is very little discussed in the manuscript, only information about patients’ gender, age and family status is presented.

2) Conclusions are too short and limited to rather general statements; its is not discussed how the research results could improve clinical practices and care standards for HD patients.

Author Response

Dear reviewer

Thank you for your constructive review for the paper. here are some feedback for your comments. 

Thanks and regards. 
